# Transcriptomic Analysis Reveals the Regulatory Mechanism of Cold Tolerance in *Saussurea involucrata*: The Gene Expression and Function Characterization of Dehydrins

**DOI:** 10.3390/ijms26189030

**Published:** 2025-09-17

**Authors:** Tongyao Chen, Lisi Zhou, Jun Zhu, Shunxing Guo, Chengcheng Liu, Airong Wang, Xu Zeng, Xiaomei Chen

**Affiliations:** 1State Key Laboratory of Bioactive Substance and Function of Natural Medicines, Institute of Medicinal Plant Development, Chinese Academy of Medical Sciences and Peking Union Medical College, Beijing 100193, China; cty930618@163.com (T.C.);; 2Xinjiang Institute of Materia Medica, Urumqi 830010, China

**Keywords:** cold stress, F_2_SK_2_ dehydrin, K-segment, RNA-seq, abiotic stresses

## Abstract

*Saussurea involucrata*, a rare and endangered medicinal plant of the Asteraceae family, is primarily distributed in high-altitude rocky slopes and meadows at elevations of 2400–4100 m. In nature, this herb endures various abiotic stresses, including intense cold and ultraviolet radiation. In our study, transcriptomic profiles revealed that most of the differentially expressed genes (DEGs) enriched in stress response pathways, such as “response to water”, “response to abscisic acid”, “cold acclimation”, and “response to water deprivation”, were significantly upregulated after low-temperature treatment. In contrast, the majority of genes with lower expression were related to “photosynthesis”, “protein–chromophore linkage”, and “chloroplast thylakoid membrane”. Among them, Kyoto Encyclopedia of Genes and Genomes (KEGG) and Gene Ontology (GO) database analysis revealed that approximately 20 DEGs were identified and annotated as dehydrin genes (*DHNs*). Quantitative PCR (qPCR) validation also confirmed that these *DHNs* were upregulated under cold stress. Moreover, *SiDHN3*, a new dehydrin gene, was cloned by Rapid Amplification of cDNA Ends (RACE). *SiDHN3*’s heterologous expression in *E. coli* showed enhanced salt, osmotic, freeze–thaw, and cold stress tolerance. A functional analysis of *SiDHN3*’s truncated derivatives revealed that the K-segment was critical for its protective function under freeze–thaw and cold stresses. Collectively, our study demonstrated the potential role of various *DHNs* as a functional protein, enhancing tolerance to cold stress in the high-altitude adaptation of plants.

## 1. Introduction

In the Asteraceae family, *Saussurea involucrata* Kar. et Kir. has a long history of ethnic medicinal use with high economic value. Many studies have shown that this herb may have various pharmacological activities, such as anti-inflammatory and analgesic effects, as well as anti-rheumatoid arthritic, anti-obesity, and anti-cancer effects [1,2,3,4,5]. In nature, *S. involucrata* is a rare and endangered plant with limited resources due to harsh living conditions, excessive harvesting, and immature artificial cultivation techniques. The habitat is mainly distributed in high mountain rocky beaches, meadows, and other areas ranging from 2400 to 4100 m [6]. Low temperatures, low oxygen, and high ultraviolet radiation levels are major unfavorable conditions for *S. involucrata* survival. Chilled temperatures (0 °C–12 °C) can inhibit the growth and development of most plants [7], while some alpine plants in extreme habitats, such as *Saussurea* species, have evolved many survival and reproduction strategies for resistance against these low temperatures [8,9]. However, research on the molecular mechanisms of adapting to low temperatures in *S. involucrata* is limited. At present, only some aquaporin proteins and dehydrins in *S. involucrata* have been verified in tobacco and tomato for their cold tolerance function [10,11,12,13].

Dehydrins, a subgroup of the late embryogenesis abundant (LEA) protein family, are important in the response to abiotic stress in plants [7,14] and has received widespread attention recently [15,16,17]. When plants encounter low temperatures and drought, *DHNs* accumulate extensively [18]; for example, *DHN5* in barley can be significantly upregulated after low-temperature treatment [19]. In addition, cold-resistant plants exhibited higher expression levels of *DHNs* than sensitive phenotypes when exposed to low temperatures [20]. In recent years, research progress on cold resistance has shown that dehydrins in plants can protect and restore the activity of lactate dehydrogenase (LDH) and catalase in freeze–thaw conditions [21,22,23], scavenge free radicals by binding to their easily oxidized amino acid sequences [22], and increase the expression levels of ROS-related genes [24,25]. Meanwhile, functional studies have also suggested that dehydrins can respond to hormones such as abscisic acid, salicylic acid, and jasmonic acid to coordinate plant development and stress responses [26]. However, little research has been carried out on dehydrins in alpine plants such as *S. involucrata*.

In our study, the morphological and physiological profiles of *S. involucrata* seedlings treated with different low temperatures were investigated. Transcriptomic analysis characterized the classification and expression pattern of dehydrin genes (*DHNs*) in *S. involucrata*, determining their responses to varying degrees of low temperature. A new differentially expressed dehydrin gene, *SiDHN3*, was cloned. Subsequently, a prokaryotic expression system in *Escherichia coli* was established, expressing a recombinant protein. The protective role of *SiDHN3* against multiple abiotic stresses, including freeze–thaw cycles, low-temperature exposure, and high osmotic pressure, was investigated.

Our study lays the groundwork for deciphering the survival strategies of *S. involucrata* in extreme habitats while providing insights into alpine plant stress response mechanisms. Furthermore, it advances our understanding of dehydrins in extremophytes, paving the way for deeper exploration of the plant dehydrin family.

## 2. Results

### 2.1. Morphological Profiles of S. involucrata at Low Temperature

Compared with the control group, there were no significant morphological changes observed in the cold or chilled groups of *S. involucrata* seedlings (Figure 1A–C), including the length and number of roots or leaves (Figure 1E). In contrast, the fresh weight in the cold group (170.07 ± 47.68 mg) and chilled group (167.80 ± 34.53 mg) treated with low temperatures was significantly reduced compared to the control group (281.07 ± 13.32 mg).

### 2.2. Physiological Profiles of S. involucrata at Low Temperature

As shown in Figure 1D, malondialdehyde (MDA), soluble protein, and soluble sugar in *S. involucrata* seedlings in each group were measured. Among them, the MDA (33.35 ± 4.70 nmol/g) and soluble protein content (0.39 ± 0.14%) in the chilled group were significantly higher than those in the control group (21.72 ± 3.38 nmol/g and 0.18 ± 0.07%). However, there was no significant difference in MDA, soluble protein, and soluble sugar between the cold and control groups. The results indicated that plant membrane lipids were subjected to low-temperature stress and underwent peroxidation after 4 °C/0 °C treatment.

### 2.3. RNA-Seq Profiles of S. involucrata at Low Temperature

The RNA-seq datasets are available in the BioProject (Accession Number: PRJNA1033840) repository of the National Center for Biotechnology Information (NCBI). Since there were no reference genomes of *S. involucrata*, the clean reads from all libraries were pooled together and de novo assembled into transcripts. A total of 87,143 unigenes were obtained to analyze sample differences and screen differentially expressed genes (DEGs, |log2(FoldChange)| > 2, q-value < 0.05). A Venn plot (Figure 2A) revealed that there were only 1436 and 4293 DEGs in the cold and chilled groups, indicating various response mechanisms in *S. involucrata* at different low temperatures. Furthermore, compared with the control group, 3141 and 2163 genes were significantly up- and downregulated in the cold group, and the numbers in the chilled group were 4739 and 3510.

### 2.4. Annotation of S. involucrata Transcripts

Gene annotation was applied by BLAST against the NCBI Non-redundant Protein Sequence (Nr)/Nucleotide Sequence (Nt), Protein families (Pfam), Swiss-Prot Protein Knowledgebase (Swiss-Prot), Gene Ontology (GO), and Kyoto Encyclopedia of Genes and Genomes (KEGG) databases. In total, there were 72,818 unigenes (83.56% of all unigenes) annotated in at least one database. Among them, a total of 68,624 unigenes (78.75%) were annotated in the Nr database. Subsequently, DEGs with Fragments Per Kilobase of transcript per Million mapped fragments (FPKM) > 100 were annotated to the top 20 terms of GO classification (Figure 2D,E). Compared with the control group, DEGs in the cold group were significantly enriched in “response to water”, “response to abscisic acid”, “cold acclimation”, and “response to water deprivation”. All DEGs in these four pathways were upregulated (Figure 2D). It indicates that the low-temperature response system of *S. involucrata* has been activated under the condition of 4 °C. Moreover, the GO terms of DEGs in the chilled group were also significantly enriched in the cold-related pathways, including “response to water”, “response to abscisic acid”, “cold acclimation”, etc. (Figure 2E). However, “photosynthesis”, “light harvesting”, “Chlorophyll binding”, and “Photosystem I” were the top five GO terms for DEGs only in the chilled vs. control group, suggesting that *S. involucrata* photosynthesis was significantly inhibited.

### 2.5. Expression Analysis of DEGs Related to Cold Stress

Based on all of unigenes, there were 51, 79, and 25 genes annotated to “response to water”, “cold acclimation”, and “response to water deprivation”. Of the 131 unigenes annotated to these three pathways, a total of 20 DEGs were annotated as proteins in the dehydrin subgroup (Table 1). Among them, there were five, two, four, and three unigenes annotated as embryogenic cell protein 40-like(ECP 40-like), dehydrin Xero 1-like, dehydrin Rab18-like, and dehydrin ERD14-like genes assigned with *Cynara cardunculus* var. *scolymus*. Additionally, eleven unigenes were annotated to three species of dehydrin (*S. involucrata*, *Artemisia annua*, and *C. cardunculus* var. *scolymus*). Compared with the control group (Table 1), five ECP 40-like, two Xero 1-like, one Rab18-like, three ERD14-like, and seven dehydrin genes were significantly upregulated in the cold group; four, two, one, three, and eight of these genes were distinctly expressed higher in the chilled group.

### 2.6. qPCR Analysis of Putative DHNs

As illustrated in Figure 2F, we performed quantitative PCR analysis on 10 selected *DHNs* putatively. The expression of these *DHNs* was low in the control group, while it was increased in the cold and chilled group. This result was consistent with the transcriptome profiles, indicating that our experimental data were reliable.

### 2.7. DHN Gene Expression in Different Periods of Treatment at Multiple Low Temperatures

As shown in Figure 3, most of the *DHNs* were more significantly upregulated in the chilled than in the cold group, except Xero-1 like (CL1854.Contig7) and dehydrin (CL9026.Contig1 and CL9026.Contig5).

As shown in Figure 3A, the expression of each *DHNs* was detected at 4 °C for day 0 (control), day 1, day 3, day 7, and day 14 (4 °C for day 7 and 20 °C for day 7). Indeed, the highest expression of almost all *DHNs* appeared on the seventh day, except CL227.Contig10 (on the first day) and CL227.Contig14 (on the third day). However, *DHNs* expression was different when the temperature was 4 °C/0 °C (day/night). According to Figure 3B, the highest expression of many *DHNs* appeared on the first or third day such as CL227.Contig10, CL1767.Contig3, CL9026.Contig1, etc. Significantly, all of the *DHNs* were distinctly downregulated in the day 14 group, suggesting that these genes were induced by low temperatures and reversed under 20 °C.

### 2.8. Sequencing Analysis of SiDHN3

The nucleotide sequence of the *SiDHN3* ORF was translated into amino acids using SnapGene 7.1.2, homologous sequences were aligned using Clustal Omega, and the alignment results were visualized in Jalview 2.11.4.0. *SiDHN3* was classified as an F_2_SK_2_-type dehydrin based on its conserved motifs (Figure 4).

### 2.9. Expression and Protective Effect of SiDHN3 Proteins on E. coli

The molecular weight of *SiDHN3* was predicted using ExPASy ProtParam to be approximately 48.593 kDa. A prominent ~50 kDa protein band was exclusively detected by SDS-PAGE in *E. coli* harboring the pET32a-*SiDHN3* plasmid, while the empty vector control showed no signal at this position, confirming successful expression (Figure 5B).

To simulate high-salinity and osmotic stress conditions, varying concentrations of NaCl and mannitol were supplemented into LB medium. The results revealed that under high concentrations of 750 mM NaCl and 750 mM mannitol, the accumulation of *SiDHN3* exerted a statistically profound enhancement on *E. coli* survival (*p* < 0.01). Notably, when exposed to osmotic stress with 1.0 M mannitol in LB medium, *E. coli* expressing *SiDHN3* exhibited a statistically significant survival advantage compared to controls (*p* < 0.05).

Following three freeze–thaw cycles, *SiDHN3* demonstrated a statistically significant cryoprotective effect on *E. coli* compared to the control group (*p* < 0.05).

When subjected to low-temperature stress at −10 °C and −20 °C with varying exposure durations (12 h and 24 h), *E. coli* expressing *SiDHN3* demonstrated significant cryoprotective effects. As shown in Figure 5A,C, the growth profiles of *E. coli* were consistent with the statistical results of CFU, which indicated that *SiDHN3* could protect *E. coli* from various stress conditions.

### 2.10. Functional Validation of SiDHN3 Protective Domains

The growth profiles of *E. coli* expressing truncated *SiDHN3* variants under freeze–thaw cycles and low-temperature stress are presented in Figure 6. Simultaneous deletion of both K-segments (∆K) abolished the cytoprotective function of *SiDHN3*, rendering transformed cells susceptible to both freeze–thaw stress and sustained low-temperature exposure. During 60 s freeze–thaw cycles, deletion of either the F-segment (∆F) or S-segment (∆S) significantly compromised (*p* < 0.05) SiDHN3-mediated protection, though to a lesser extent than ∆K (Figure 6B).

Under 12 h low-temperature treatments, F-segment deletion (∆F) alone reduced *E. coli* survival (*p* < 0.05) across tested temperatures (−10 °C to −20 °C), indicating its essential role in freeze–thaw stress mitigation (Figure 6B).

## 3. Discussion

Plants frequently delay growth as a strategy to adapt to low-temperature stress [27]. This response is particularly pronounced in alpine species; for example, *S. leontodontoides* exhibits inverse correlations between altitude and vegetative organ biomass [28]. Consistent with these observations, *S. involucrata* showed significantly reduced fresh weight under low-temperature conditions, though root length, root number, and leaf characteristics remained unaffected. Concurrently, levels of osmoregulatory compounds—specifically MDA and soluble proteins—increased significantly in chill-treated (4 °C/0 °C) plants relative to controls.

Low temperatures typically disrupt water absorption–transpiration equilibrium, inducing cellular dehydration. Consequently, proteins related to water deprivation like dehydrins are known to accumulate rapidly under cold stress and are widely studied [26,29,30]. Supporting this, GO analysis revealed significant enrichment of cold-related terms (“response to water”, “cold acclimation”, and “response to water deprivation”) among DEGs in 4 °C/4 °C and 4 °C/0 °C groups. Twenty of these DEGs encoded annotated dehydrin proteins.

Substantial *DHN* upregulation during cold exposure is well-documented in pepper (*Capsicum* spp.), *Solanum sogarandinum*, and barley (*Hordeum vulgare*) [18,19,31,32]. Cold-tolerant plants exhibit enhanced *DHNs* expression [20], exemplified by higher *DHN1*/*DHN2* transcript levels in winter barley cultivars than in spring genotypes [15,19]. Furthermore, in both wheat and barley, highly tolerant varieties exhibit a more rapid and substantial accumulation of dehydrins compared to less tolerant varieties [33]. Critically, the heterologous expression of cold-adapted *DHNs* enhances low-temperature tolerance in recipient plants [34,35], as demonstrated by improved cold resistance in transgenic tomato and tobacco expressing *S. involucrata*-derived *SiDHN2* [36] or *SiDHN* [12]. Our data align with this paradigm: *DHN* expression in *S. involucrata* was cold-inducible and stress intensity-dependent. Notably, virtually all *DHNs* were markedly downregulated after 7-day recovery at 20 °C, indicating reversible cold responsiveness, a pattern also observed for *RcDHN1-5* in *Rhododendron catawbiense* [37].

Despite the established importance of dehydrins, functional studies on *S. involucrata* remain scarce, with only two genes characterized to date. This contrasts sharply with the extensive structural and functional knowledge of dehydrins in other species [30], where they mitigate diverse abiotic stresses including salinity [38,39], drought [40,41], and cold [42].

Here, we isolated the full-length coding sequence of a new cold-induced dehydrin (CL1767.Contig1) via RACE amplification. The protein, designated as *SiDHN3*, contains two F-segments, one S-segment, and two K-segments, classifying it as the first documented F_2_SK_2_-type dehydrin from *S. involucrata*. Heterologous expression in *E. coli* significantly enhanced bacterial survival under low temperatures, freeze–thaw cycles, and hyperosmotic stress.

While dehydrins are classically categorized into five types (K_n_, SK_n_, K_n_S, Y_n_K_n_, and Y_n_SK_n_) by conserved domains [43,44], expanding sequence/functional data reveal limitations in predicting properties based solely on Y-/S-/K-segment counts [26]. Their complex regulation necessitates segment-specific functional analyses. The K-segment, a core motif implicated in cellular protection, antioxidation, and protein/membrane stabilization [45,46,47,48,49], mediates macromolecular interactions (e.g., binding acidic phospholipid vesicles and membrane domains) [45,50]. Truncation analysis confirmed the K-segment’s indispensability for *SiDHN3*-mediated multistress protection in *E. coli*, with efficacy scaling with the K-segment copy number.

*SiDHN3* represents the first cloned F-segment containing dehydrin from *S. involucrata*. This conserved phenylalanine pair-containing motif [51] remains poorly characterized but may facilitate macromolecule binding and cold-labile enzyme stabilization [52]. Crucially, F-segment deletion significantly reduced freeze–thaw stress protection by *SiDHN3* in *E. coli* assays.

As proteins responsive to multiple stresses, dehydrins play various protective roles in plant cells [53], mechanisms that are closely linked to their structures [54]. Further investigation into the structure and function of *SiDHN3* in model organisms is crucial, particularly given its significant promise for application in economic crops. Meanwhile, elucidating the subcellular localization of dehydrins can help clarify their functional mechanisms [55]. For instance, dehydrins localized to the plasma membrane often exhibit membrane cryoprotection functions [56,57]. Additionally, *S. involucrata*, adapted to high-altitude environments, is a valuable model for studying dehydrins. Genomic studies of its DHNs [58] could clarify their ecological role in adaptation.

In summary, *S. involucrata* showed significantly reduced fresh weight under low-temperature conditions; transcriptomic profiles revealed that most of the differentially expressed genes (DEGs) enriched in the stress response pathways were significantly upregulated after low-temperature treatment. Among them, KEGG and GO analysis revealed approximately 20 DEGs and annotated them as *DHNs*. The coding sequence of *SiDHN3*, a differentially expressed *DHNs*, was cloned, representing the first characterized F_2_SK_2_-type dehydrin in *S. involucrata*. *SiDHN3* significantly enhanced *E. coli* survival under diverse abiotic stresses, including low temperatures and freeze–thaw cycles. The protective function depended on the K-segment within the structure of *SiDHN3*, while the F-segment proved critical for freeze–thaw stress tolerance. These findings elucidate molecular adaptations underlying the high-altitude resilience *S. involucrata* and advances our understanding of dehydrin-mediated abiotic stress protection in plants.

## 4. Materials and Methods

### 4.1. Plant Materials

*S. involucrata* was germinated in 1/2MS medium at 20 °C in a Ningbo Jiangnan DRXM incubator (Ningbo Jiangnan DRXM, Ningbo, China) in our laboratory. Based on the methodology previously described [59], the 30-day-old seedlings were divided into three groups for treatment at different temperatures for 7 days, including the control (20 °C), cold (4 °C), and chilled groups (4 °C/0 °C, day/night).

At the end of 7 days, five repetitions were collected from each group. The fresh weight and length of the leaves and roots were measured, as well as the number of roots and leaves. The same samples were immersed in liquid nitrogen for rapid freezing and were restored at −80 °C for physiological profiles and RNA-seq.

### 4.2. Determining the Physiological Characteristics of S. involucrata

Referring to the manufacturer’s instructions for the malondialdehyde (MDA) assay kit and plant soluble sugar content test kit (Nanjing Jiancheng Bioengineering Institute, Nanjing, China), as well as the Bradford Protein Assay Kit (Beyotime Biotech, Shanghai, China), the MDA, soluble sugar, and total soluble protein content of *S. involucrata* were detected.

### 4.3. Transcriptomic Analysis

Sample processing, library construction, sequencing, assembly, and analysis were completed by BGI Tech Solutions Co., Ltd. (Shenzhen, China). In brief, the raw data was filtered with SOAPnuke (v1.4.0) to obtain clean reads [60]. After removing reads containing adapters, as well as ensuring that the unknown base (“N” base) ratio is more than 5% and the low-quality base ratio is more than 20%, clean reads were assembled by Trinity (v2.0.6) and BUSCO (v5.7.0) to assess the assembly quality [61]. Then, clean reads were mapped to unigenes by Bowtie2 (v2.2.5), and RSEM (v1.2.8) was used to calculate the expression of unigenes [62,63]. Distinct unigenes were annotated with NCBI Nr/Nt, Pfam, Swiss-Prot, GO, and KEGG databases.

PossionDis was performed for between-group differential expression analysis, using |log2(FoldChange)| > 2, q-value < 0.05, and PFKM > 100 as a threshold to identify differentially expressed genes (DEGs). An advanced Circos bar plot was plotted by an online platform (https://www.bioinformatics.com.cn, accessed on 1 November 2023).

### 4.4. Expression of DHNs from Different Groups

In order to verify the reliability of the RNA-seq data, samples from the same libraries were used for quantitative PCR analyses. The total RNA of each sample was extracted using the Quick RNA isolation Kit (Huayueyang Biotechnology, Beijing, China), and the cDNA library was obtained by using HiScript III All in one RT SuperMix (Vazyme, Nanjing, China). In total, 1 μL of cDNA diluted with 10 μL of ddH_2_O was used as the template.

Primers for qRT-PCR (Appendix A) were designed with Primer Premier software (Primer Premier v5.0; Premier Biosoft International, Palo Alto, CA, USA). The expression level of the target genes was determined by qTR-PCR using the ROCHE LightCycler^®^ 480 II system (Roche, Basel, Switzerland) and Taq Pro Universal SYBR qPCR Master Mix (Vazyme, Nanjing, China) in 15 μL reactions. Each reaction consisted of 5 ng of total RNA, 0.3 μL of each primer, and 7.5 μL master mixes. In total, we performed three biological replicates and three technical replicates. The PCR reactions were performed in a thermocycler with the following conditions: 5 min at 95 °C, 45 cycles of 10 s at 95 °C, 10 s at 60 °C, and 10 s at 72 °C. The *GAPDH* gene was used as a reference [59]. The 2^−∆∆Ct^ method was used for evaluating gene expression [64].

### 4.5. Expression of DHNs in Different Periods of Treatment at Multiple Low Temperatures

In plants, *DHNs* can be constantly strengthened with a longer duration in the cold or lower temperature stress. For investigating *DHN* expression patterns, we performed quantitative PCR analysis on 10 candidate genes in different periods at multiple low temperatures. Here, *S. involucrata* seedlings were germinated for 30 days. As described in a previous study with slight adjustments [65], for cold treatment, samples from day 0, day 1 (4 °C), day 3 (4 °C), day 7 (4 °C), and day 14 (7 day 4 °C + 7 day 20 °C) groups were collected. For the chilled treatment, the same procedure was carried out at 4 °C/0 °C. Next, qPCR was performed according to the method described in Section 4.4.

### 4.6. Cloning of Full-Length Coding Sequence of SiDHN3

Total RNA was extracted from *S. involucrata* using the Quick RNA isolation Kit (Huayueyang Biotechnology, Beijing, China). The 5′- and 3′-cDNA library construction and full-length coding sequence of *SiDHN3* were obtained by the 5′- and 3′- rapid amplification of cNDA ends (RACE) using the SMARTerTM RACE cDNA Amplification kit (Clontech, California, CA, USA) according to the manufacturer’s instructions.

### 4.7. Sequence Analysis of SiDHN3

The open reading frames (ORFs) of *SiDHN3* were identified using the ORF Finder tool (https://www.ncbi.nlm.nih.gov/orffinder/, accessed on 3 June 2024). The amino acid sequences were compared with reported plant dehydrin sequences through the NCBI BLAST Program (https://blast.ncbi.nlm.nih.gov/Blast.cgi, accessed on 3 June 2024), and high-similarity protein matches were performed for multiple sequence alignments with Clustal Omega (https://www.ebi.ac.uk/jdispatcher/msa/clustalo, accessed on 30 March 2025). The phylogenetic tree was generated using MEGA 11.0.13 [66].

### 4.8. Construction and Expression of SiDHN3 and Its Truncated Derivatives

Truncated derivatives (∆F1, ∆F2, ∆S, ∆K1, ∆K2) were generated by sequentially removing segments. Derivative constructs ∆F and ∆K were created by deleting two F-segments and two K-segments, respectively (the sequence design of the *SiDHN3*-truncated derivative is shown in Appendix A). The plasmids were engineered to contain the *SiDHN3*-truncated derivatives by GenScript (Suzhou, China). The recombinant plasmids (pET32a-*SiDHN3*, ∆F1, ∆F2, ∆S, ∆K1, ∆K2, ∆F, ∆K, and the control vector pET32a; the pET32a-*SiDHN3* plasmid diagram is shown in Appendix A) were transformed into an *E. coli* strain OrigamiB (DE3) (Beyotime, Shanghai, China) according to Beyotime’s protocol. The sequencing results were assembled with Codoncode Aligner (Codoncode, Centerville, MA, USA).

### 4.9. Abiotic Stress Tolerance Assays

*E. coli* OrigamiB (DE3) cells carrying pET32a-*SiDHN3* plasmids and pET32a (control) were grown in liquid LB medium at 37 °C and 180 rpm until an OD600 of 0.5–0.6 was reached followed by 20 h of induction at 16 °C and 150 rpm with 0.2 mM IPTG. The OD600 of the cell cultures was then adjusted to 0.5, and they were then diluted serially to 10-3, 10-4, 10-5, and 10-6 with LB. Next, 5 μL of each sample was spotted separately on LB plates containing NaCl (500 mM, 750 mM) and mannitol (500 mM, 750 mM, 1.0 M) for salinity and osmotic treatments. For the freeze–thaw test, the OD-adjusted cells were placed in liquid nitrogen for 30 s and 60 s separately and thawed at room temperature; after freeze–thaw repetitions, the cultures were diluted and spotted on the control LB plate. All plates were incubated at 37 °C for 24 h, and colony forming units (CFUs) were counted at 10^−5^ or 10^−6^ dilution [58]. For the statistical analysis of colony forming units (CFUs), an independent-samples *t*-test was performed to compare differences between two groups. A one-way ANOVA followed by Tukey’s post hoc test was used for comparisons among multiple groups.

## Figures and Tables

**Figure 1 ijms-26-09030-f001:**
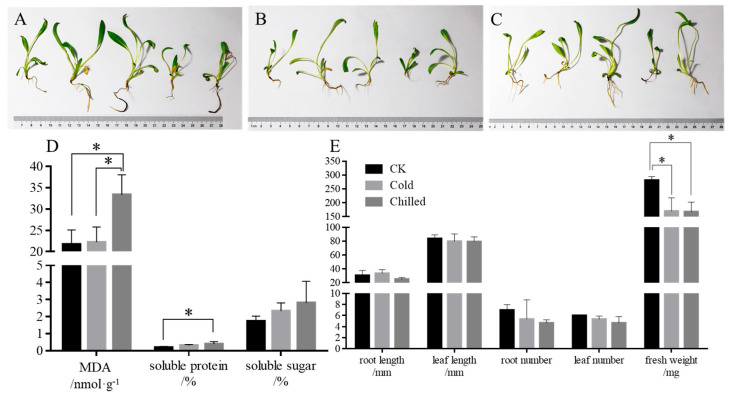
Morphological and physiological profiles of *S. involucrata*. (**A**–**C**) The phenotypes of the control (**A**), cold (**B**), and chilled (**C**) group; (**D**) the content of osmoregulatory substances; *: statistically significant, *p* < 0.05, *N* = 3. The color of each bar corresponds to the categories defined in the legend of (**E**). (**E**) The growth index of roots, leaves, and fresh weight. *: statistically significant, *p* < 0.05, *N* = 5. MDA: malondialdehyde.

**Figure 2 ijms-26-09030-f002:**
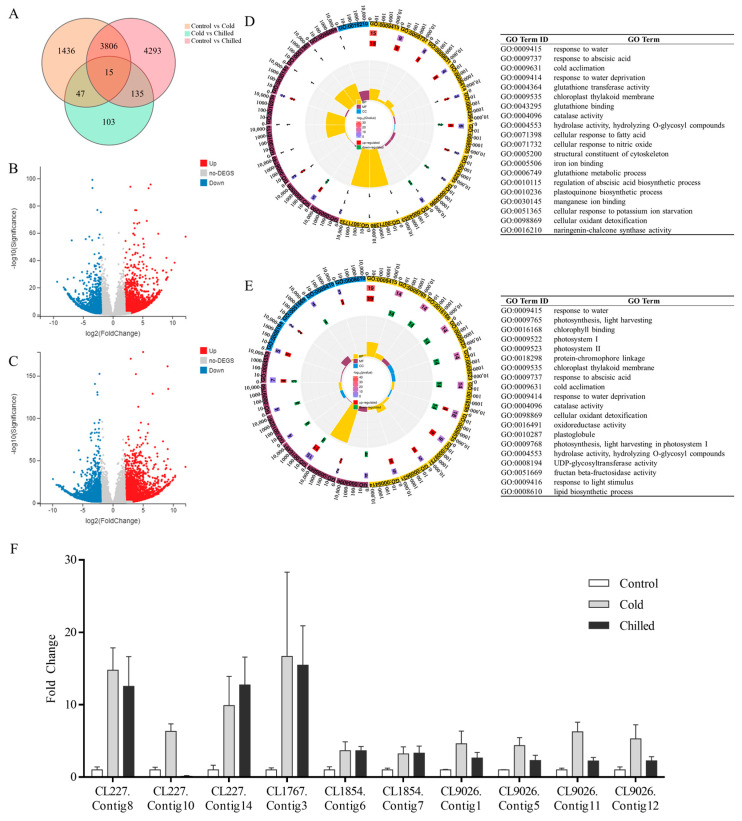
Transcriptome profiles of *S. involucrata* under low temperatures. (**A**) Venn diagram of *S. involucrata* in response to various low-temperature stress treatments. (**B**) Volcano plot of cold vs. control. (**C**) Volcano plot of chilled vs. control. (**D**) Circos bar plot of Gene Ontology (GO) database. classification for DEGs between cold and control. (**E**) Circos barplot of GO classification for DEGs between chilled and control. (**F**) Expression analysis of putative DEGs from RNA-seq. *N* = 3.

**Figure 3 ijms-26-09030-f003:**
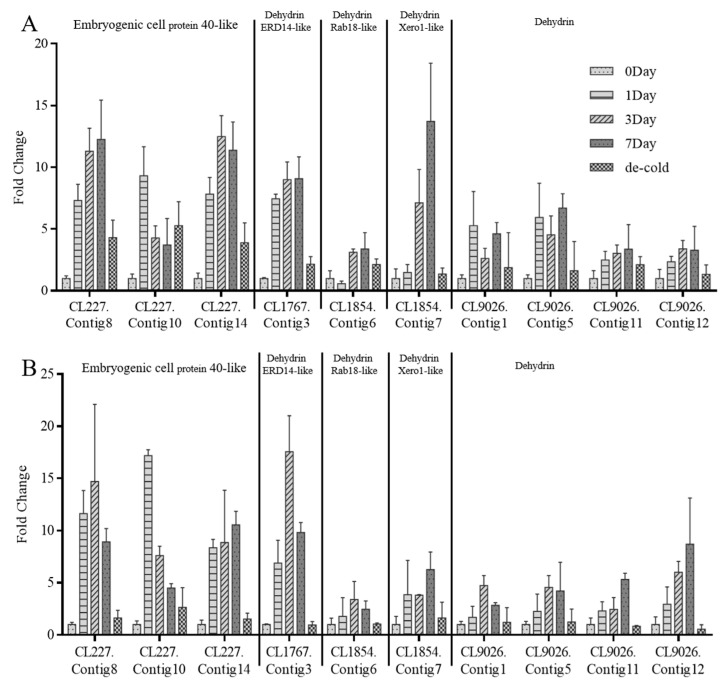
Expression analysis of putative *DHNs* from *S. involucrate* in different periods. (**A**) *DHNs* expression in seedlings treated at 4 °C/4 °C in different periods. (**B**) *DHNs* expression in seedlings treated at 4 °C/0 °C in different periods. *N* = 3.

**Figure 4 ijms-26-09030-f004:**
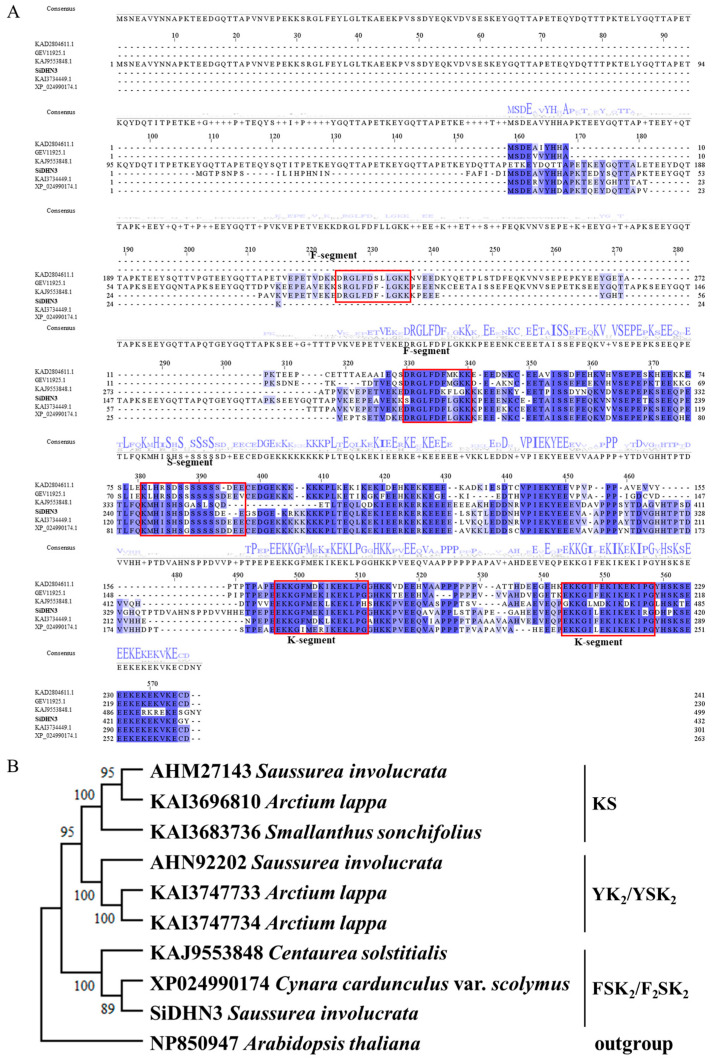
Analysis of *SiDHN3* amino acid sequence characteristics. (**A**) Multiple sequence alignment profile of *SiDHN3* amino acid with homologs. (**B**) Phylogenetic analysis of *SiDHN3* and homologous *SiDHN* amino acid sequences. Blue letters indicate the consensus sequence from the multiple sequence alignment. The red box highlights the identified dehydrin segment.

**Figure 5 ijms-26-09030-f005:**
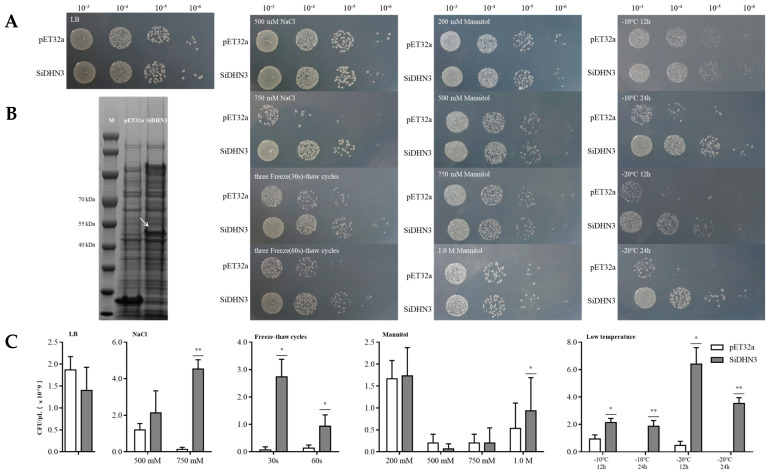
Expression analysis and functional validation of *SiDHN3*. (**A**) Growth profile of pET32a and pET32a-*SiDHN3* strains in various stress environments. (**B**) Recombinant *SiDHN3* expression profiling. (**C**) CFU of pET32a and pET32a-*SiDHN3* strains subjected to various stresses. *: compared with the control group (pET32a), *p* < 0.05; **: compared with the control group (pET32a), *p* < 0.01. *N* = 3. The arrow points to the band corresponding to the target protein, *SiDHN3*. LB: Luria-Bertani agar. M: ColorMixed Protein Marker 10–180 kDa.

**Figure 6 ijms-26-09030-f006:**
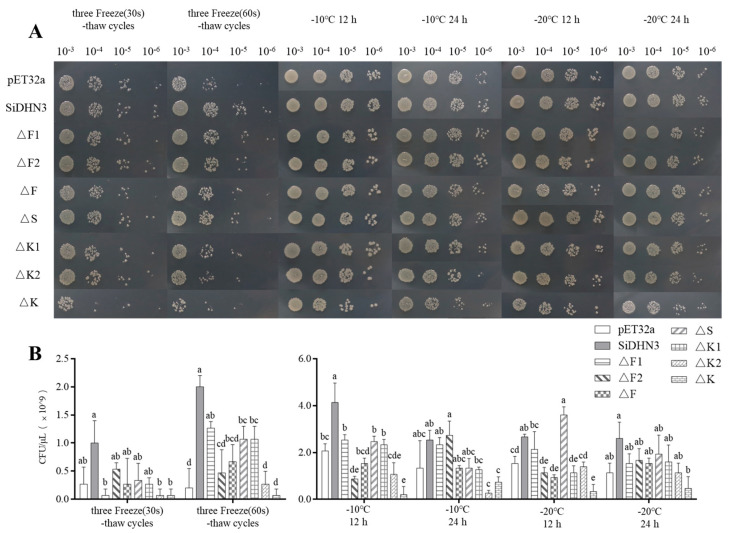
Functional validation of *SiDHN3*-truncated derivatives. (**A**) Growth profile of *SiDHN3*-truncated derivative strains in various stress environments. (**B**) CFU of *SiDHN3*-truncated derivative strains subjected to various stresses. Different lowercase letters indicate significant differences between different habitats at the 0.05 level. ∆F1, ∆F2, ∆S, ∆K1, and ∆K2 are the truncated derivatives that were generated by sequentially removing segments.

**Table 1 ijms-26-09030-t001:** Expression of dehydrin genes.

No.	Gene id	Gene	Reference Species	log2 (Cold/Control)	log2 (Chilled/Control)
1	CL227.Contig10	embryogenic cell protein 40-like	*C. cardunculus* var. *scolymus*	5.45	5.09
2	CL227.Contig4			5.76	5.11
3	CL227.Contig8			4.22	3.95
4	CL227.Contig14			3.88	3.13
5	CL227.Contig6			2.98	1.93
6	CL1854.Contig4	dehydrin Xero 1-like		3.63	4.02
7	CL1854.Contig7			3.58	4.00
8	CL1854.Contig6	dehydrin Rab18-like		2.91	2.95
9	CL1767.Contig3	dehydrin ERD14-like		5.40	6.00
10	CL1767.Contig1			3.62	4.23
11	CL1767.Contig4			2.73	3.57
12	CL9026.Contig6	dehydrin	*S. involucrata*	6.01	5.78
13	CL9026.Contig4			3.09	3.44
14	CL9026.Contig10			1.70	3.19
15	CL9026.Contig5			1.54	3.11
16	CL9026.Contig11			3.07	3.10
17	CL9026.Contig1			2.14	2.27
18	CL9026.Contig8			4.88	6.11
19	CL9026.Contig12			2.34	2.72
20	CL9026.Contig7			2.11	1.40

## Data Availability

The raw data supporting the conclusions of this article will be made available by the authors without undue reservation.

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
