# Peer review of "Transcriptomic Analysis Reveals the Regulatory Mechanism of Cold Tolerance in Saussurea involucrata: The Gene Expression and Function Characterization of Dehydrins"

_ijms, 2025, doi:10.3390/ijms26189030_

Round 1

Reviewer 1 Report

Comments and Suggestions for Authors

This study focuses on the alpine medicinal plant Saussurea involucrata and systematically investigates its molecular responses to cold stress through transcriptome sequencing and functional validation. The results demonstrated that multiple stress-related pathways were significantly activated, with approximately 20 differentially expressed genes annotated as dehydrins (DHNs). qPCR confirmed their cold-induced expression patterns, and a novel gene, SiDHN3 (F2SK2 type), was successfully cloned. Functional assays showed that SiDHN3 markedly enhanced the survival of E. coli under salt, osmotic, cold, and freeze–thaw stresses. Truncation analysis further revealed that the K-segment is indispensable for protection against low-temperature and freeze–thaw stresses, while the F-segment also plays a critical role in freeze–thaw tolerance. Overall, this work deepens our understanding of cold adaptation in alpine plants and provides new evidence for the functional roles of dehydrins.

However, I have several suggestions for further improvement:

  1. It is recommended to overexpress or silence SiDHN3 in model plants (e.g., Arabidopsis thaliana or tobacco) and evaluate survival rate, photosynthetic efficiency, and electrolyte leakage to better validate its in planta function.

  2. Construct a SiDHN3–GFP fusion protein to determine its subcellular localization, which may provide insights into its functional mechanism.

  3. Assess ROS accumulation and antioxidant enzyme activities in transgenic materials under cold stress to clarify whether SiDHN3 enhances cold tolerance via ROS scavenging.

  4. Compare SiDHN3 expression and cold tolerance among S. involucrata populations from different altitudes or habitats to support its ecological role in natural adaptation.

Author Response

Dear Reviewer,

Thank you for your comments and suggestions concerning our manuscript entitled “Transcriptomic analysis reveals the regulatory mechanism of cold tolerance in Saussurea involucrata: the gene expression and function characterization of dehydrins” (ID: ijms-3852665). These suggestions are all valuable and very helpful for revising and improving our paper, as well as the important guiding significance to our work. We have studied suggestions carefully and have made revise which we hope meet with approval. Revised portion are clearly highlighted using the “Track Changes” function in the paper. The revise in the paper and the reply to the reviewer’s suggestions are as follows:

1) Reply to suggestion for further improvement: “It is recommended to overexpress or silence SiDHN3 in model plants (e.g., Arabidopsis thaliana or tobacco) and evaluate survival rate, photosynthetic efficiency, and electrolyte leakage to better validate its in planta function.”

Reply: We fully agree with your insightful suggestion regarding the direction of our future research. Functional studies of SiDHN3 in model plants are indeed essential for a comprehensive understanding of its biological roles. However, due to the relatively long growth cycles of model plants, such functional experiments typically require at least 6 to 12 months to complete. Therefore, we are unable to incorporate these results into the current manuscript in a timely manner. Nonetheless, this line of investigation will remain a priority in our subsequent work on SiDHN3.

In response to your valuable recommendation, we are pleased to include a prospective outlook in the Discussion section (Lines 297–300):” As proteins responsive to multiple stresses, dehydrins play various protective roles in plant cells [53], mechanisms that are closely linked to their structures [54]. Further investigation into the structure and function of SiDHN3 in model organisms is crucial, particularly given its significant promise for application in economic crops.”

2) Reply to suggestion for further improvement: “Construct a SiDHN3–GFP fusion protein to determine its subcellular localization, which may provide insights into its functional mechanism.”

Reply: Elucidating the subcellular localization of SiDHN3 is essential for understanding its protective mechanisms under stress in S. involucrata. However, the experimental materials required for subcellular localization analysis require an extended growth period. Currently, our laboratory lacks suitable plant materials at the appropriate developmental stage to conduct this investigation promptly. Nevertheless, the subcellular localization of SiDHN3 will be included as an important component of our future research.

In response to your valuable recommendation, we are pleased to include a prospective outlook in the Discussion section (Lines 300–303): “Meanwhile, elucidating the subcellular localization of dehydrins can help clarify their functional mechanisms [55]. For instance, dehydrins localized to the plasma membrane often exhibit membrane cryoprotection functions [56, 57].”

3) Reply to suggestion for further improvement: “Assess ROS accumulation and antioxidant enzyme activities in transgenic materials under cold stress to clarify whether SiDHN3 enhances cold tolerance via ROS scavenging.”

Reply: This suggestion is highly valuable for elucidating the mechanism by which SiDHN3 confers stress protection. The oxidative stress response and the activity of antioxidant enzyme systems represent key mechanisms through which plants respond to various abiotic stresses. Dehydrins in numerous species have been demonstrated to protect antioxidant enzyme activities under such conditions.

In our subsequent studies, we will investigate the effect of SiDHN3 expression in model plants on ROS accumulation and antioxidant enzyme activities. Additionally, we plan to conduct in vitro assays to verify the protective role of SiDHN3 on antioxidant enzyme activities under low-temperature stress. Through these complementary approaches, we aim to further explore the functional role of SiDHN3.

4) Reply to suggestion for further improvement: “Compare SiDHN3 expression and cold tolerance among S. involucrata populations from different altitudes or habitats to support its ecological role in natural adaptation.”

Reply: In fact, S. involucrata (snow lotus) is classified as a national second-class protected plant in China. Wild populations are legally protected, yet their habitats continue to shrink due to human activities and climate change. Under laboratory conditions, we are currently only able to cultivate seedlings, which cannot be overwintered to mature into perennial plants.

This suggestion is highly meaningful for advancing research on dehydrins in this species; however, the current conservation status and cultivation limitations make it unfeasible to obtain samples from different altitudes or geographic origins. Nonetheless, I am confident that as research on S. involucrata deepens, breakthroughs will be made in its cultivation and conservation. We will continue to investigate the ecological roles of dehydrins in its natural adaptation processes.

Once again, thank you very much for your comments and suggestions.

Note: “Line Number” of reviewers referred to many sentences have been changed in revised manuscript because of revisions were made and clearly highlighted using the “Track Changes” function in Microsoft Word

Reviewer 2 Report

Comments and Suggestions for Authors

Dear Authors,

Reviewer comments ijms-3852665

The manuscript entitled „Transcriptomic analysis reveals the regulátory mechanism of cold tolerance Saussurea involucrata: the gene expression and function characterization of dehydrins“ represents a useful study aimed at an investigation of the structure, sequence analysis, and biological functions of SiDHN3 protein and its truncated derivatives with deleted F1, F2, S, K1, and K2 segments on abiotic stress tolerance using E. coli assays using pET32a-SiDHN3 plasmid transformants. The results of transcriptomic analysis in cold-treated Saussurea involucrata are validated by qPCR on 10 putative dehydrins. The manuscript thus provides a complex view on the structure and biological functions of dehydrins isolated from Saussurea involucrata.

I can recommend the manuscript for publication in International Journal of Molecular Sciences.

However, I have a few minor comments on the present manuscript.

1/ In Materials and methods, the source of Saussurea plants used for the experiments has to be specified, i.e., from which institution or locattion the plants were obtained. Analogously, the source of E. coli and the structure or a relevant reference on the pET32a-SiDHN3 plasmid used for E. coli transformation have to be given.

2/ In Materials and methods as well as in Results, Figure 6 legend, the kind of statistical test used for the determination of significant differences has to be specified.

3/ In Discussion, the third paragraph on DHN upregulation under cold and increased DHN expression in cold-tolerant winter barley cultivars iwth respect to spring ones I would recommend the authors to cite also papers by Vítámvás et al. (2019) Front. Plant Sci. 10, 7. https://doi.org/10.3389/fpls.2019.00007 on differential dehydrin protein accumualtion in differentially tolerant winter wheat and barley cultivars overwintering in the field, and to cite a review by Kosová et al. (2021) Plants 10,789. https://doi.org/10.3390/plants10040789

on dehydrins and their functions in cold-treated wheat and barley plants.

4/ Terminology and formal comments on the text:

Materials and methods, line 283: The 4/0 °C (day/night) treatment is cold treatment, not freezing treatment!! Freezing is defined as temperatures below zero.

Materials and methods, line 314: Correct the term „qRT-PCR“ (not „(qTR-PCR“)!!

Discussion, line 218: Use the term „altitude“ instead of „elevation“!!

Results, Line 117: Modify the word form „significant“ to „significantly“ in the statement: „Moreover, the GO terms of DEGs in the freezing group were also significantly enriched…“¨

Line 129: Remove the second verb „were“ in the statement: „Based on all of unigenes, there were 51, 79 and 25 genes annotated to „response to water“…“

Line 157: Modify the statemnet as follows: „Significantly, all of DHNs wer distinctly down-regulated at 14 day group…“

Lines 204, 365: Add a space between a number and a corresponding unit in „30 s“ and „60 s“.

Discussion, line 226: Add the verb „are“ in the statement. „Consequently, proteins related to water deprivation like dehydrins are known to accumulate rapidly under cold stress and are widely studied….“

Final recommendation: Accept after a minor revision.

Author Response

Dear Reviewer,

Thank you for your comments and suggestions concerning our manuscript entitled “Transcriptomic analysis reveals the regulatory mechanism of cold tolerance in Saussurea involucrata: the gene expression and function characterization of dehydrins” (ID: ijms-3852665). These suggestions are all valuable and very helpful for revising and improving our paper, as well as the important guiding significance to our work. We have studied suggestions carefully and have made revise which we hope meet with approval. Revised portion are clearly highlighted using the “Track Changes” function in the paper. The revise in the paper and the reply to the reviewer’s suggestions are as follows:

1) Reply to suggestion: “In Materials and methods, the source of Saussurea plants used for the experiments has to be specified, i.e., from which institution or location the plants were obtained. Analogously, the source of E. coli and the structure or a relevant reference on the pET32a-SiDHN3 plasmid used for E. coli transformation have to be given.”

Reply: (1) The S. involucrata plants used in this study were sterile seedlings cultivated under laboratory conditions. To clarify the origin of the plant materials, the following sentence has been added to Section “4.1. Plant material” (Line 334):

S. involucrata was germinated in 1/2MS medium at 20℃ at Ningbo Jiangnan DRXM incubator (Ningbo Jiangnan DRXM, China) in our laboratory.”

(2) We have specified the source of the E. coli strain in line 417: " … …was transformed into E. coli strain OrigamiB (DE3) (Beyotime, Shanghai)”.

(3) A schematic diagram of the pET32a-SiDHN3 plasmid construct is provided in Supplementary Figure S2.

Figure S2 Schematic diagram of the pET32a-SiDHN3 plasmid

2) Reply to suggestion: “In Materials and methods as well as in Results, Figure 6 legend, the kind of statistical test used for the determination of significant differences has to be specified.”

Reply: As described in the Methods section (4.9 Abiotic stress tolerance assays), the statistical analyses pertaining to Figure 6 are now provided in lines 431-436: " For the statistical analysis of colony forming units (CFUs), an independent-samples t-test was performed to compare differences between two groups. A one-way ANOVA followed by Tukey's b post hoc test was used for comparisons among multiple groups".

3) Reply to suggestion: “In Discussion, the third paragraph on DHN upregulation under cold and increased DHN expression in cold-tolerant winter barley cultivar  s with respect to spring ones I would recommend the authors to cite also papers by Vítámvás et al. (2019) Front. Plant Sci. 10, 7. https://doi.org/10.3389/fpls.2019.00007 on differential dehydrin protein accumualtion in differentially tolerant winter wheat and barley cultivars overwintering in the field, and to cite a review by Kosová et al. (2021) Plants 10,789. https://doi.org/10.3390/plants10040789

on dehydrins and their functions in cold-treated wheat and barley plants.”

Reply: We sincerely thank you for suggesting these two references, which help broaden our international literature coverage. As recommended, we have incorporated Kosová et al. (2021) as reference [32] at line 255, and Vítámvás et al. (2019) as reference [33] in the Discussion section. The corresponding text has been added at lines 258–259:“Furthermore, in both wheat and barley, highly tolerant varieties exhibit more rapid and substantial accumulation of dehydrins compared to less tolerant varieties [33].”.

4) Reply to suggestion: “Terminology and formal comments on the text: Materials and methods, line 283: The 4/0 °C (day/night) treatment is cold treatment, not freezing treatment!! Freezing is defined as temperatures below zero.”

Reply: Given that freezing is strictly defined as temperatures below 0 °C, and to clearly distinguish between the 4 °C and 4 °C/0 °C treatments, we have revised the manuscript by renaming the "4 °C/0 °C" group as the "chilling group". This change has been consistently applied throughout the text, as well as in all figures and tables.

5) Reply to suggestion: “Terminology and formal comments on the text: Materials and methods, line 314: Correct the term „qRT-PCR“ (not „(qTR-PCR“)!!

Discussion, line 218: Use the term „altitude“ instead of „elevation“!!

Results, Line 117: Modify the word form „significant“ to „significantly“ in the statement: „Moreover, the GO terms of DEGs in the freezing group were also significantly enriched…“¨

Line 129: Remove the second verb „were“ in the statement: „Based on all of unigenes, there were 51, 79 and 25 genes annotated to „response to water“…“

Line 157: Modify the statemnet as follows: „Significantly, all of DHNs wer distinctly down-regulated at 14 day group…“

Lines 204, 365: Add a space between a number and a corresponding unit in „30 s“ and „60 s“.

Discussion, line 226: Add the verb „are“ in the statement. „Consequently, proteins related to water deprivation like dehydrins are known to accumulate rapidly under cold stress and are widely studied….“”

Reply: Thank you very much for your thorough review and valuable corrections. We have carefully addressed each of your comments and have implemented all modifications using the “Track Changes” mode in Microsoft Word for clear visibility throughout the manuscript.

Once again, thank you very much for your comments and suggestions.

Note: “Line Number” of reviewers referred to many sentences have been changed in revised manuscript because of revisions were made and clearly highlighted using the “Track Changes” function in Microsoft Word

Reviewer 3 Report

Comments and Suggestions for Authors

The paper (Chen et al., entitled “Transcriptomic analysis reveals the regulatory mechanism of cold tolerance in Saussurea involucrata: the gene expression and function characterization of dehydrins) submitted for review is technically sound, has a scientific and practical interest. In general, the study was conducted at an adequate methodological level. However, some questions and comments require clarification.

Avoid using keywords that are listed in the article title. The keywords must be revised.

The Introduction and Discussion sections should be enhanced by analysing more publications published during the previous five years (2020+). Moreover, these sections mostly rely on the results of Chinese scientists. However, this article was submitted to an international journal. This means that a comprehensive analysis of the problem must be presented, drawing from both Chinese and international scientific publications.

Add research limitations to the Discussion section.

Add perspectives for future exploration to the Discussion section.

Author Response

Dear Reviewer,

Thank you for your comments and suggestions concerning our manuscript entitled “Transcriptomic analysis reveals the regulatory mechanism of cold tolerance in Saussurea involucrata: the gene expression and function characterization of dehydrins” (ID: ijms-3852665). These suggestions are all valuable and very helpful for revising and improving our paper, as well as the important guiding significance to our work. We have studied suggestions carefully and have made revise which we hope meet with approval. Revised portion are clearly highlighted using the “Track Changes” function in the paper. The revise in the paper and the reply to the reviewer’s suggestions are as follows:

1) Reply to suggestion: “Avoid using keywords that are listed in the article title. The keywords must be revised.”

Reply: To optimize the precision of literature retrieval, the keywords "Saussurea involucrata", "dehydrins", and "cold tolerance", which duplicated those in the title, have been strategically replaced with " Cold stress ", "F2SK2 dehydrin", and " K-segment ".

2) Reply to suggestion: “The Introduction and Discussion sections should be enhanced by analysing more publications published during the previous five years (2020+). Moreover, these sections mostly rely on the results of Chinese scientists. However, this article was submitted to an international journal. This means that a comprehensive analysis of the problem must be presented, drawing from both Chinese and international scientific publications.”

Reply: Saussurea involucrata is a rare and endangered plant species primarily found in the Tianshan Mountains in Xinjiang, China, with minor populations distributed in Russia and Kazakhstan. As a result, Chinese scholars have contributed the largest body of research on this species, which is reflected in the references cited. Given that dehydrins are a widely distributed protein family across plant species globally, we have incorporated nine additional international scientific publications as references ([32–33], [39], [53–58]) into the Discussion section, eight of which were published after 2020.

3) Reply to suggestion: “Add research limitations to the Discussion section. Add perspectives for future exploration to the Discussion section”

Reply: The following text has been added to the Discussion section (lines 297–305) as a subsection on study limitations and future directions:” As proteins responsive to multiple stresses, dehydrins play various protective roles in plant cells [53], mechanisms that are closely linked to their structures [54]. Further investigation into the structure and function of SiDHN3 in model organisms is crucial, particularly given its significant promise for application in economic crops. Meanwhile, elucidating the subcellular localization of dehydrins can help clarify their functional mechanisms [55]. For instance, dehydrins localized to the plasma mem-brane often exhibit membrane cryoprotection functions [56, 57]. Additionally, S. involucrata, adapted to high-altitude environments, is a valuable model for studying dehydrins. Genomic studies of its DHNs [58] could clarify their ecological role in adaptation.”

Once again, thank you very much for your comments and suggestions.

Note: “Line Number” of reviewers referred to many sentences have been changed in revised manuscript because of revisions were made and clearly highlighted using the “Track Changes” function in Microsoft Word

Round 2

Reviewer 1 Report

Comments and Suggestions for Authors

Accept